# Organic vs. Non-Organic Plant-Based Foods—A Comparative Study on Phenolic Content and Antioxidant Capacity

**DOI:** 10.3390/plants12010183

**Published:** 2023-01-01

**Authors:** Álvaro Cruz-Carrión, Ma. Josefina Ruiz de Azua, Begoña Muguerza, Miquel Mulero, Francisca Isabel Bravo, Anna Arola-Arnal, Manuel Suarez

**Affiliations:** Nutrigenomics Research Group, Departament de Bioquímica i Biotecnologia, Universitat Rovira i Virgili, 43007 Tarragona, Spain

**Keywords:** agriculture, farming, fruit, nuts, polyphenols, vegetables

## Abstract

A plant’s stress response involves the production of phytochemicals, including phenolic compounds. Their synthesis can be modulated by organic (ORG) or non-organic (NORG) farming systems in which they are grown. To examine this issue, thirteen plant-based foods cultivated in ORG and NORG systems were compared in terms of antioxidant capacity, total content of phenolics, anthocyanins, flavan-3-ols and flavonols. The results showed that NORG fruits tended to have higher phenolic compounds content, whereas ORG fruits had more antioxidant capacity. NORG legume stood out for having higher values from all the parameters analyzed in comparison to its ORG equivalent. ORG nuts showed more flavan-3-ols and flavonols than their NORG counterparts, nonetheless, tended to be less antioxidant. ORG vegetables displayed higher phenolics and anthocyanins, which reflected in higher antioxidant capacity than NORG ones. These findings suggest that farming systems differentially modulate phenolic compound composition and antioxidant capacity based on the plant species studied.

## 1. Introduction

Studies show that consumers are now inclined to choose fruits and vegetables that are healthier and produced by a more sustainable and environmentally friendly agricultural system [1]. According to the Council of the European Union, organic farming respects rules based on fundamental principles, such as prohibition of the use of genetically modified organisms (GMOs); prohibition on the use of ionizing radiation; and limitation of the use of artificial fertilizers, herbicides and pesticides [2]. Recent production and market trends show how important organic products have become over the last decade. In the last ten years, organic farming has risen by more than 70%, indicating a very dynamic and quickly rising sector [3]. In fact, the organic agriculture is practiced in 187 countries and more than 106 billion euros were sold globally in 2019 in the organic food and drink sector [4]. This increase in the production of plant-based organic foods can be attributed to the consumer’s preference for these products, possibly characterized by better taste and often produced in areas close to the place of consumption. In addition, it is supposed that they contain a higher content of beneficial and health-promoting substances [5]. However, when choosing between organic and non-organic products, consumers have doubts about the nutritional quality [5]. In this sense, a diet rich in fruits and vegetables has been associated with reducing the incidence of chronic diseases, owing to the bioactive compounds they contain, notably phenolic compounds [6]. Specifically, studies have provided evidence that consumption of phenol-rich foods generates protective effects against chronic diseases, including cardiovascular diseases, neurodegeneration and cancer [7,8,9]. The phenolic compounds are the most predominant bioactive compounds in the diet, reaching values of up to 1 g/day [6]. Structurally, phenolic compounds can be classified as flavonoids and non-flavonoids; in turn, the main subclasses of flavonoids are anthocyanidins, flavan-3-ols, flavanones, flavones, flavonols and isoflavones, while among the non-flavonoids of nutritional relevance are phenolic acids [9]. In nature, phenolic compounds can be found in a wide range of dietary components and medicinal plants such as tea, fruits and vegetables. In particular, the profile of phenolic compounds present in plants differs among species [10,11]. The synthesis of phenolic compounds in plants can be influenced by environmental conditions such as water availability, soil fertilization and the mineral content of the soil. In particular, the phenolic profile of plant-based foods is known to differ according to systems of farm management, i.e., organic vs. non-organic [12]. Although it has not yet been proven that organic products contain the highest content of nutrients, they should certainly contain the lowest content of pesticides and substances harmful to health, such as nitrates [5]. Information supporting both agricultural practices can be obtained from the literature [6,13,14]. However, the results of the research are inconclusive, and there is insufficient evidence to affirm the differences in nutritional value between organically and non-organically grown plant-based foods. Based on this, it is plausible to hypothesize that agriculture practices, for instance non-organic farming (NORG) and organic farming (ORG), generate a differential effect on phenolic synthesis and antioxidant activity among plant-based foods. To explore this issue, the aim of this study was to evaluate the effects of ORG or NORG farming management systems on the antioxidant capacity and phenolic compounds content in plant-based foods grown in Tarragona, Spain.

## 2. Results

In this study, we selected thirteen plant-based foods from the same cultivar, which were organically and non-organically cultivated in Tarragona, Spain. The total content of anthocyanins, flavan-3-ols, flavonols, phenolics and antioxidant capacity were determined in each of the plant-based foods to elucidate whether non-organic farming (NORG) generates the same influence on the synthesis of these compounds and antioxidant activity as organic farming (ORG).

### 2.1. Phenolic Compounds Content in Organic and Non-Organic Plant-Based Foods

The phenolic compounds content (PCC) in ORG and NORG plant-based foods can be observed in Figure 1, which shows that the thirteen plant-based foods from two farming systems evaluated in this study had a specific phenolic signature. Regarding the TPC (Figure 1a), there is a pattern of higher TPC in NORG plant-based foods, with higher values in 6 of the 9 samples. Figure 1b shows that the total anthocyanins content in plant-based foods was detected in 6 of the 13 samples, of which 67% varied statistically, although no trend in favor of either farming type could be stated. As expected, these compounds were not detected in any of the nuts or carob pod under analysis and in vegetables it was only detected in organic sweet pepper. Moreover, 54% of the plant-based foods showed differences on total flavan-3-ols content (TFaC) between ORG and NORG (Figure 1c), while the rest exhibited similar contents. Among those that differed statistically, 4 of 7 plant-based foods with the highest flavan-3-ols content were found to be NORG. Finally, the total flavonols content (TFoC) between ORG and NORG was statistically different in 9 of the 13 plant-based foods evaluated (Figure 1d). Here, 56% of plant-based foods with the highest TFoC values were found to be ORG.

The data in Figure 2 show the distribution of the variability in PCC (i.e., TPC, TAC, TFaC and TFoC) from ORG plant-based foods compared to its NORG counterparts. It was found that 44% of the plant foods showed no variation between ORG and NORG. Among the PPCs that varied statistically between the ORG and NORG, 33% of the differences in TPC favored organically grown plant-based foods. The same trend was observed in 50% of the products in TAC, 48% in TFaC and 44% in TFoC. In brief, 41% of ORG plant-based foods were more abundant in the PCC. However, when considering all the differences as a whole, an average percentage difference in the abundance of organic samples was calculated for each PCC in question showing that TPC is −6%; TAC is +6%; TFaC is +7%; and TFoC is +20%. Interestingly, the organic farming was, on average, more abundant in all the phenolic fractions determined, except TPC.

### 2.2. Antioxidant Capacity of Organic and Non-Organic Plant-Based Foods

The antioxidant capacity of ORG and NORG plant-based foods determined by ORAC assay is displayed in Table 1. The ORAC values ranged from 336.0 to 6565.5 µmol Trolox Eq/100 g fw in ORG samples, while in NORG samples ranged from 338.0 to 5627.8 µmol Trolox Eq/100 g fw. Significant differences of the effects of farming systems on the antioxidant capacity were evident in 8 of the 13 samples studied, although a clear tendency favoring one of them was not observed, with half higher in the ORG samples and half in the NORG samples. Similarly, the data in Figure 3 represent the distribution of variability in the significant ORAC values from ORG plant-based foods compared to NORG.

No differences were found in the antioxidant capacity of olive and tomato E. from ORG and NORG. Tomato T. grown non-organically had 1.2-fold higher ORAC values than their ORG equivalent. Moreover, organically grown oranges and sweet cherry had ORAC values 1.2-fold higher than their NORG equivalents. On the other hand, Ekstasis tomato showed the lowest ORAC levels among samples of both types of farming (ORG, 336.0 and NORG, 338.0 µmol Trolox Eq/100 g fw).

Among the vegetables studied, onion and sweet pepper showed similar antioxidant capacity between organically and non-organically grown samples. However, Swiss chard had 1.3-fold higher ORAC levels than its NORG counterpart.

Almond, hazelnut H. and carob pods grown non-organically had 1.2-fold higher ORAC values than their ORG equivalents, while ORG hazelnut C. showed 1.1-fold higher ORAC values than its NORG analogous. Moreover, carob pods were the plant-based food with the highest ORAC value among ORG (6565.5 µmol Trolox Eq/100 g fw) and NORG (5627.8 µmol Trolox Eq/100 g fw).

### 2.3. Multivariate Data Analysis

Simultaneously, all variables, including TPC, TAC, TFaC, TFoC and ORAC, were used to perform the unsupervised method called principal component analysis (PCA), after plant-based foods data were preprocessed by autoscaling. PCA was used to find the directions that best explain the variance in plant-based foods without referring to class labels. Figure 4 illustrates the PCA results between ORG and NORG groups, including fruits (Figure 4a), vegetables (Figure 4b), and nuts and legume (Figure 4c), in 3D score plots of the first three principal components (PCs), and PCA biplot between PC1 and PC2, showing the direction of the impact caused by the different parameters under study.

The PCA score plot of the fruit comparison presented a 99% cumulative variance explained of the first three PCs, with no clear patterns of separation. PCA results showed that the principal component (PC1; 60.7%) separates the data mainly due to TAC followed by TFaC levels, while PC2 (33.1%) separates the data with respect to TPC and ORAC, and PC3 (5.2%) shares the same separating variables as PC2 (Figure 4a).

On the other hand, the PCA score plot for the comparison between ORG and NORG vegetables showed clear patterns of separation for all samples, as illustrated in Figure 4b. The first three PCs accounted for 91.9% of the cumulative variance explained, with principal component 1 (PC1; 58.8% of the variance) separating the data mainly by ORAC levels, followed by TFaC, while PC2 (21.9% of the variance) separates the data by ORAC levels, followed by flavan-3-ols content; and PC3 (11.2% of the variance) divides the vegetables mainly by ORAC and flavonols content. This suggests that the most important factor in differentiating organically or non-organically grown vegetables is the ORAC value. It was observed that the data points corresponding to sample 8, i.e., Swiss chard, were the most distant from each other, namely ORG (−0.9, 1.7, 0.5 as distances to the origin) and NORG (−0.7, 0.6, −0.6 as distances to the origin), showing a strong impact of the farming system, indeed, it was the only vegetable that varied in ORAC levels between the two types of farming systems. In addition, the PC3 scores (11.2%) of the ORG and NORG vegetables were positive and negative, respectively. This result suggests that the PC3 score was positively related to the ORG farming system. In this case, ORG vegetables that have large positive factor loadings in PC3 tend to increase TPC, TAC, TFoC and ORAC, and those with negative factor loadings tend to decrease TFaC in this type of cultivation.

The PCA score plot of the comparison between ORG and NORG nuts and legume showed that the principal component (PC1; 72.7%) separates the data mainly due to ORAC values and flavan-3-ols content, whereas PC2 (18.0%) separates the data mainly with respect to flavonols content and ORAC values, and PC3 (8.6% of the variance) divides nuts and legume mainly by flavan-3-ols and flavonols content (Figure 4c), with no clear pattern of separation.

The PCs are determined by the contribution of all plant-based foods. The impact of a distinct plant-based food on the principal component is captured in its loading value. The plots of the PCA loadings (Figure 4a–c) display the contribution of single plant-based food to the global separation seen among samples. It becomes obvious from these plots that the differences are mainly driven by a small number of plant-based foods. Interestingly, the shared plant-based foods with the highest loading plot scores are almost identical between the three groups: TAC and TFaC for fruits group, ORAC and TFaC for vegetables group, and ORAC and TFaC for nuts and legume, and are influencing PC1 in the families of the plant-based foods.

## 3. Discussion

The results obtained reflect the phenolic content of a group of plant-based foods that have been harvested in the different seasons of the year (spring 2018–winter 2019). In this line, work by Young et al. [15] studied the phenolic levels in leaf lettuce (*Lactuca sativa* L. cv. Kalura and Red Sails), collards (*Brassica oleracea* L. cv. Top Bunch), and pac choi (*Brassica rapa* L. cv. Mei Qing) cultivated under ORG and NORG during the summer 2003. Another study evaluated the phenolic acids concentration in wheat harvested in spring and winter 2008 [16]. Likewise, Ponder and Hallmann [17] studied the effects of ORG and NORG farming systems on the polyphenol content in three raspberry cultivars collected in the summer 2013 and in the autumn 2014. The results found indicate that most of the plant-based foods analyzed presented significant differences in total phenolic content, with a trend in favor of non-organic farming. The same trend was found in the study by Letaief et al. [18] in orange and orange juice. Similarly, the results obtained by Barrett et al. [19] demonstrated that tomato juice prepared from organically produced tomatoes on four farms was significantly lower in total phenolics content than conventional tomatoes, and these results were significantly different among specific growers. On the contrary, Tarozzi et al. [20] reported that organic red oranges cv. Tarocco have a higher phenolic content than integrated red oranges. In this vein, it has also been reported that organic tomatoes cv. Redondo were richer in total phenolics (+24%) than conventional ones [21]. Similarly, a study by Faller and Fialho [6] reported that organic fruits tend to have higher hydrolysable polyphenol contents than conventional ones. These different results may be because the fruits analyzed differ between their study and ours, only one coincides. In addition, it is known that farming practices gives rise to distinct patterns of phenolic profiles according to the plant species studied, with some species more susceptible to the induction of phenolic compounds synthesis than others [6]. This can be verified in the current research, for which non-organic fruits exhibited higher total phenolic content, while in the case of vegetables it was exhibited by the organic ones.

There is scientific evidence for a correlation between a soil-fertilization-dependent effect and flavonoid content [12]. In this way, more than half of the non-organically grown plant-based foods studied were notable for having higher total flavan-3-ols and flavonols contents than their organic counterparts. However, overall, the ORG samples exhibited 7% and 20% greater abundance of flavan-3-ols and flavonols, respectively, than NORG ones. Thus, nitrogen provision can be based on inorganic and/or organic soil fertilization, and the reduction is theoretically possible due to the biochemical pathway that drives the synthesis of flavonoids [12]. Similarly, Lima et al. [22] informed that total content of flavonoids was higher in most of the analyzed conventional vegetables (zucchini squash, banana, potato, eggplant, orange, lime, mango, passion fruit and radish), possibly because of farming practice adopted. These findings are in line with the results obtained on plums, where flavonols were considerably higher in conventionally farmed fruits [23]. However, differences in flavonols content because of varietal differences cannot be excluded, because flavonoid contents may show wide variations in different varieties of fruits and vegetables [5]. Indeed, in our study, non-organic tomatoes cv. Ekstasis and Tores stood out for their higher phenolic content than their organic counterparts, while Caris-Veyrat et al. [24] reported that organic tomatoes from three different varieties (Félicia, Izabella and Paola) had higher polyphenol content than the conventional counterparts. As stated previously, the ORG tomato T. varied by −100% in terms of TAC and −56% in terms of TFaC with respect to its NORG counterpart, suggesting that anthocyanins content generates the greatest differentiation between organic and non-organic fruits. Concerning the total anthocyanins content, although no clear overall trend was observed in plant-based foods when comparing the two farming systems, on average, the organic samples proved to be 6% more abundant in anthocyanins than the non-organic samples. In this same way, other studies also identified no statistical differences in the content of phenolic compounds in relation to the type of farming system, such as a study comparing two eggplant cultivars [25] and another comparing strawberry [26]. As mentioned above, the phenolic compounds content is known to be highly dependent on the cultivar and farming conditions [12]. In our work, agrometeorological parameters such as relative humidity, daily temperature, scalar wind speed, accumulated daily precipitation, daily global solar radiation, reference evapotranspiration and altitude were collected and analyzed, without finding any significant correlation or significant effect on phenolic content or antioxidant activity (Appendix A).

Variation in phenolic antioxidant capacity is closely related to the class and concentration of phenolic compounds [1]. Our results showed that as a function of the higher abundance of phenolic content in ORG fruits and vegetables, antioxidant compounds tend to increase in them. As it has been reported, synthesis of these compounds is enhanced in response to phytopathogenic infections, in accordance with their proposed role in plant defense mechanisms [5]. Infected plant tissue and resistant tissue have been identified as being characterized by a general change in metabolic pattern that includes the activation of phenol-oxidizing enzymes and peroxidases [27]. In this line, Wang and Millner [28] concluded that organic farming led to an improvement in antioxidant activity in blueberries due to higher content of phenolic acids and anthocyanins compared to a non-organic farming. Likewise, Stracke et al. [29], found that organic apples exhibited on average 15% more antioxidant content than non-organic fruits. Most authors agree that the type of farming system can influence the phytochemical composition of the plants and therefore, by implication, the level of antioxidant activity [5]. It has been described that organic farming has potential to influence the synthesis of antioxidants, increasing their levels and therefore increasing antioxidant capacity, due to the fact that this farming system does not supply as much nitrogen as conventional fertilizers and also causes more stress to the plants than non-organic farming [5].

Among the plant-based foods studied, carob pods, both NORG and ORG, stood out for presenting the highest values of TPC, TFaC and ORAC. Additionally, NORG carob pods also exhibited the highest TFoC, coinciding with what has previously been reported that carob pods contain considerable amounts of phenolic compounds with remarkable antioxidant properties [30]. Additionally, among the plant-based foods analyzed, orange seems to be more affected by alteration in farming system, since statistical differences were revealed between organic and non-organic oranges in all the determinations, without any clear trend in support of either farming system. Remarkably, however, the ORG orange showed higher anthocyanins content than the NORG orange, and the same pattern was evident in ORAC values. These results confirm that phenolic molecules may be important antioxidant components in explaining the observed activity. In addition, these findings are consistent with results reported by Tarozzi et al. [20], who reported that organic red oranges exhibited significantly higher total anthocyanins and total antioxidant activity than non-organic red oranges. Another case to emphasize is that of peppers, which showed the same pattern as oranges, statistically higher TAC, numerically higher TPC and TFaC, and consequently, numerically higher ORAC in ORG samples, showing conformity with what was proved by Muscolo et al. [31], that organic fertilizers enhanced the synthesis of total phenols, flavonoids, and anthocyanins, along with antioxidant activities of red Topepo sweet peppers compared to those grown in unfertilized soil. Based on the results obtained for hazelnut and tomato cultivars, it appears that the type of cultivation system, organic or non-organic, has a differential influence between cultivars of the same species. Indeed, in a previous study comparing 23 broccoli cultivars, it was determined that phytochemicals studied individual compound concentrations responded differently and that the type of farming system, organic or non-organic, contributes to the variation in the concentration of theses phytochemicals [32].

## 4. Materials and Methods

### 4.1. Chemicals and Reagents

Fluorescein, Folin–Ciocalteu reagent, gallic acid, *p*-dimethylaminocinnamaldehyde (DMACA), quercetin, Trolox and (+)-catechin were purchased from Fluka/Sigma-Aldrich (Madrid, Spain). Cyanidin-3-*O*-rutinoside was acquired from PhytoLab (Vestenbergsgreuth, Germany). 2,2′-Azobis (2-methylpropionamidine) dihydrochloride was purchased from Acros Organics (Geel, Belgium). Standard compounds were individually dissolved in acetone/Milli-Q water/acetic acid (70/29.5/0.5; *v/v/v*) and stored at −20 °C. All standard stock solutions were newly prepared every three months.

### 4.2. Plant-Based Foods Samples

Samples of fruits: olive (*Olea europaea* L. cv. Arbequina), orange (*Citrus sinensis* L. cv. Navel), sweet cherry (*Prunus avium* L. cv. Burlat) and tomato (*Lycopersicon esculentum Mill*. cv. Ekstasis and Tores); vegetables: onion (*Allium cepa* L. var. cepa cv. Figueres), sweet pepper (*Capsicum annuum* L. cv. Italia) and Swiss chard (*Beta vulgaris var. flavescens* cv. Delta); nuts: almond (*Prunus dulcis* cv. Marcona), hazelnut (*Corylus* L. cv. Castanyera and Negreta) and walnut (*Juglans regia* L. cv. Serr); and legume: carob pods (*Ceratonia siliqua* cv. Banya de cabra) were harvested from May 2018 to February 2019 and donated by farmers and agricultural companies from Camp de Tarragona, Spain, with the exception of nuts that were kindly provided by the Institute of Agrifood Research and Technology (IRTA). Cultivars used in this study were selected by both organically (ORG) and non-organically (NORG) grown varieties available. According to the farmers, the plant-based foods were grown in strict compliance with the rules governing each of the cultivation systems [2]; indeed, the fertilizers, soil conditioners and nutrients used in organic production comply with the provisions of the current European regulations [2]. The cultures were harvested as previously described [33]. The cultures were harvested as previously described [30]. According to the information provided by them, these crops were cultivated in open fields. ORG cultivation system meets the certification provided by the Catalan Council of Organic Production. Approximately 2 kg of each plant-based food was randomly sampled, mimicking consumer purchasing behavior. The samples were washed, and the edible part was separated, chopped, frozen in liquid nitrogen and ground. Then, the samples were freeze-dried for one week in a Telstar LyoQuest freeze-dryer (Thermo Fisher Scientific, Madrid, Spain). The powders were kept at room temperature and protected from light and humidity until use. The moisture content of the fresh samples was determined by the weight loss after heating (98 °C, 24 h) [34].

### 4.3. Extraction and Quantification of Phenolic Compounds

#### 4.3.1. Extraction of Phenolic Compounds

The extraction of phenolic compounds was carried out in line with the method described by Iglesias-Carres et al. [35]. Although this method was originally optimized specifically for grapes, unpublished results in which we compared several methods confirmed that this was the most suitable extraction methodology, since most of the families of phenolic compounds present in plant-based foods are extracted. Briefly, the extraction parameters were 80 mL/g, 65% methanol (1% formic acid), 72 °C and 100 min under agitation at 500× *g*.

#### 4.3.2. Total Phenolic Content

Total phenolic content (TPC) of extracts was measured by the Folin–Ciocalteu method adapted from Nenadis et al. [36]. Briefly, 10 μL of the extract, 50 μL of Folin–Ciocalteu reagent and 500 μL of Milli-Q water were mixed and left in the dark for 3 min. Then, 100 μL of Na_2_CO_3_ (25%) was added and diluted to 1 mL with Milli-Q water. After 1 h of incubation in dark, absorbance was measured at 725 nm using an Eon BioTek spectrophotometer (Izasa, Barcelona, Spain). Gallic acid (GA) was used as standard. The results are reported as mg GA Eq/100 g fresh weight (fw).

#### 4.3.3. Total Anthocyanins Content

The total anthocyanins content (TAC) was assessed by the pH differential method [37]. Extracts were diluted with sodium acetate buffer (0.4 M, pH 4.5) and potassium chloride buffer (0.025 M, pH 1.0). Next, absorbance was read at 515 and 700 nm using an Eon BioTek spectrophotometer. TAC is expressed as milligrams of cyanidin-3-*O*-rutinoside equivalents per 100 g of fresh weight (mg Cy3R Eq/100 g fw).

#### 4.3.4. Total Flavan-3-ols Content

The total flavan-3-ols content (TFaC) of extracts was estimated by the DMACA method [38]. Concisely, 100 μL of extract samples was mixed with 500 μL of DMACA solution (0.1% 1 N HCl in methanol). After 10 min of incubation in dark, absorbance was assessed at 640 nm using an Eon BioTek spectrophotometer. Catechin concentrations were used to construct a calibration curve and TFaC values are shown as mg catechin Eq/100 g fw.

#### 4.3.5. Total Flavonols Content

The total flavonols content (TFoC) was evaluated with the method described by Cacace et al. [39]. In brief, 250 μL of extract with 250 μL of 0.1% HCl in ethanol 95% and 4.55 mL of 2% HCl were mixed and could react for 15 min. Spectrophotometric measurements at 360 nm were performed. TFoC levels, which were calculated based on the standard curve of quercetin, are reported as mg quercetin Eq/100 g fw.

### 4.4. Antioxidant Capacity

The antioxidant capacity of plant-based foods samples was determined by oxygen radical absorbance capacity (ORAC) assay described by Huang et al. [40]. Briefly, 25 μL of extract of samples was mixed with 25 μL of 73 mM 2,2′-Azobis (2-methylpropionamidine) dihydrochloride and 150 μL of 59.8 nM fluorescein. The fluorescence intensity was assessed every 2 min for 120 min using an FLx800 multidetection microplate reader (BioTek, Winooski, VT, USA; λex = 485 nm and λem = 528 nm). Trolox concentrations were used to construct a calibration curve and the results are expressed as μmol Trolox Eq/100 g fw.

### 4.5. Statistical Analysis

Student’s *t*-test (SPSS, SPSS Inc., Chicago, IL, USA) was applied to assess any differences (*p* < 0.05) in the PCC results and ORAC values between ORG and NORG plant-based foods. Results are expressed as mean ± standard deviation (SD). In addition, to holistically evaluate the impact of the agricultural system, a chemometric analysis, specifically, principal component analysis (PCA) based on normalized concentrations, was performed using the corresponding functions of MetaboAnalyst 5.0 software.

## 5. Conclusions

The farming system, organic or non-organic, generated a different content of phenolic compounds and antioxidant capacity of plant-based foods, with no clear general pattern of differentiation. Thus, it is suggested that the effects of type of cultivation tended to depend on the plant species studied and its cultivar. Specifically, the vegetable group showed a clear pattern of differentiation between the two types of farming systems, where the highest abundance of antioxidant capacity and phenolic compounds, except TFaC and TFoC, was reflected in the organic samples. Indeed, orange and tomato T. were the only two species that showed differences across all parameters, the latter showing a clear trend of higher quantity in the NORG sample. However, this study has some limitation, such as the lack of individual identification of phenolic compounds by chromatographic techniques and the limited information regarding the cultivar.

Consuming plant-based foods can promote health by providing antioxidants, and as part of a healthy diet, these plant-based foods can help lower risk factors for non-communicable diseases.

## Figures and Tables

**Figure 1 plants-12-00183-f001:**
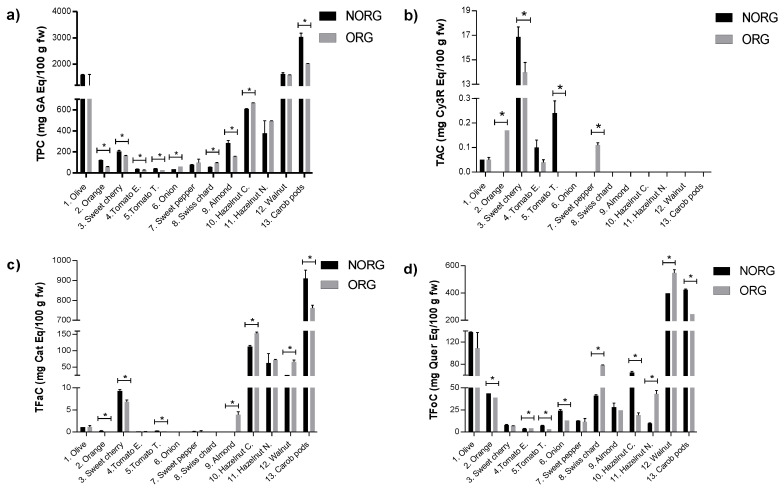
Phenolic compounds content in organic and non-organic plant-based foods: (**a**) Total phenolic content. (**b**) Total anthocyanins content. (**c**) Total flavan-3-ols content. (**d**) Total flavonols content. Samples analyzed per plant-based food (*n* = 3), analytical replicates per samples (*qc* = 3). Values are expressed as mean ± standard deviation (SD). * Statistical difference between farming practices by Student’s *t*-test (*p* < 0.05). Abbreviations: Cy3R, cyanidin-3-*O*-rutinoside; Eq, equivalent; GA, gallic acid; Hazelnut C, hazelnut cv. Castanyera; Hazelnut N, hazelnut cv. Negreta; NORG, non-organic farming; ORG, organic farming; TAC, total anthocyanins content; TFaC, total flavan-3-ols content; TFoC, total flavonols content; Tomato E, tomato cv. Ekstasis; Tomato T, tomato cv. Tores; TPC, total phenolic content.

**Figure 2 plants-12-00183-f002:**
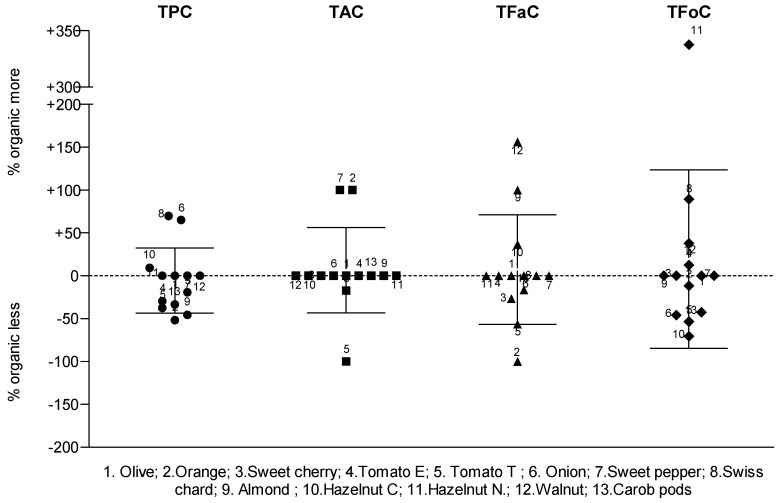
Distribution of variability in the phenolic compounds content of organic plant-based foods compared to the non-organic ones. Each point represents the % variability in PPC of the same plant-based food and cultivar, which is labeled with a number. That is, plus and minus signs refer to more or less abundance of a PPC from an organic sample with respect to non-organic sample as the baseline for comparison. Abbreviations: Cy3R, cyanidin-3-*O*-rutinoside; Eq, equivalent; GA, gallic acid; Hazelnut C, hazelnut cv. Castanyera; Hazelnut N, hazelnut cv. Negreta; NORG, non-organic farming; ORG, organic farming; PPC, phenolic compounds content; TAC, total anthocyanins content; TFaC, total flavan-3-ols content; TFoC, total flavonols content; Tomato E, tomato cv. Ekstasis; Tomato T, tomato cv. Tores; TPC, total phenolic content.

**Figure 3 plants-12-00183-f003:**
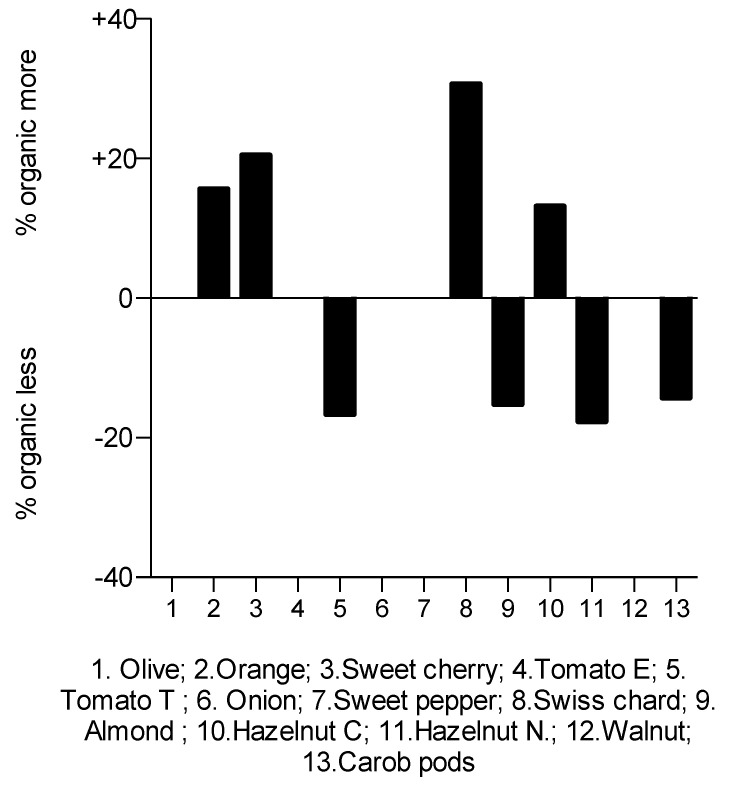
Distribution of variability in ORAC values of organic plant-based foods compared to the non-organic ones. Each point representing the % variability in ORAC values of the same plant-based food and cultivar, which is labeled with a number. That is, plus and minus signs refer to more or less abundance of an ORAC value from an organic sample with respect to non-organic sample as the baseline for comparison. Abbreviations: Hazelnut C, hazelnut cv. Castanyera; Hazelnut N, hazelnut cv. Negreta; ORAC, oxygen radical absorbance capacity; NORG, non-organic farming; ORG, organic farming; PPC, phenolic compounds content; Tomato E, tomato cv. Ekstasis; Tomato T, tomato cv. Tores.

**Figure 4 plants-12-00183-f004:**
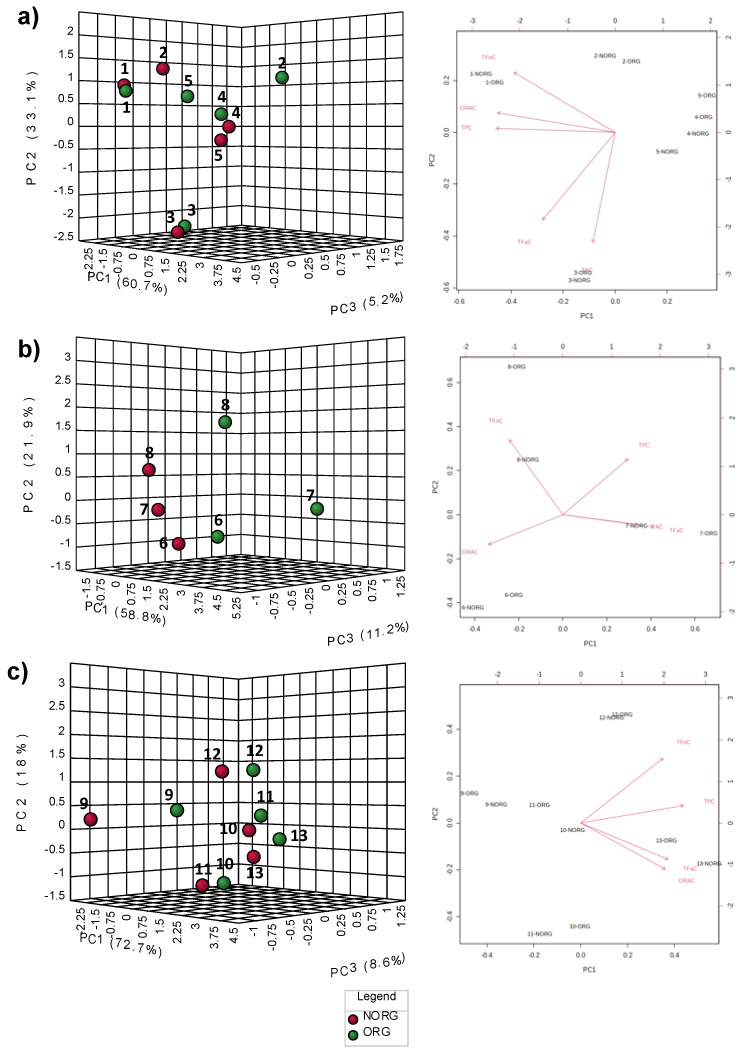
Patterns of separation between organically (ORG) and non-organically (NORG) grown plant-based foods across the three groups of samples. The figures correspond to PCA score plot: (**a**) PCA results of fruits samples. (**b**) PCA results of vegetables samples. (**c**) PCA results of nuts and legume samples. The number of data points corresponds to each sample: 1. Olive, 2. Orange, 3. Sweet cherry, 4. Tomato E., 5. Tomato T., 6. Onion, 7. Sweet pepper, 8. Swiss chard, 9. Almond, 10. Hazelnut C., 11. Hazelnut N., 12. Walnut, 13. Carob pods. Abbreviations: PCA, principal component analysis; PC, principal component.

**Table 1 plants-12-00183-t001:** Antioxidant capacity of organic and non-organic plant-based foods.

Plant-Based Foods		ORAC (µmol Trolox Eq/100 g fw)		*p* Value
		ORG	NORG	
Fruits	1. Olive	2906.1 ± 525.7	3479.8 ± 9.3	0.09
	2. Orange	994.5 ± 69.7	860.3 ± 15.5	0.03
	3. Sweet cherry	1016.2 ± 3.5	843.8 ± 79.2	0.02
	4. Tomato E.	338.0 ± 7.2	336.0 ± 1.8	0.73
	5. Tomato T.	372.2 ± 9.7	446.5 ± 1.9	0.00
Vegetables	6. Onion	691.3 ± 39.6	686.5 ± 0.1	0.88
	7. Sweet pepper	430.4 ± 42.1	410.8 ± 32.6	0.47
	8. Swiss chard	584.0 ± 0.5	447.1 ± 0.6	0.00
Nuts	9. Almond	4093.6 ± 51.1	4827.6 ± 198.8	0.00
	10. Hazelnut C.	5349.2 ± 112.4	4728.1 ± 99.6	0.00
	11. Hazelnut N.	4194.7 ± 195.6	5095.4 ± 559.7	0.03
	12. Walnut	4757.9 ± 59.7	4833.3 ± 211.2	0.58
Legume	13. Carob pods	5627.8 ± 153.7	6565.5 ± 230.8	0.00

Samples analyzed per plant-based food (*n* = 3), analytical replicates per samples (*qc* = 3). Values are expressed as mean ± standard deviation (SD). Statistical difference between agricultural practices by Student’s *t*-test (*p* < 0.05). Abbreviations: Hazelnut C, hazelnut cv. Castanyera; Hazelnut N, hazelnut cv. Negreta; NORG, non-organic farming; ORAC, oxygen radical absorbance capacity; ORG, organic farming; Tomato E, tomato cv. Ekstasis; Tomato T, tomato cv. Tores.

## Data Availability

Not applicable.

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
