# Peer review of "Organic vs. Non-Organic Plant-Based Foods—A Comparative Study on Phenolic Content and Antioxidant Capacity"

_plants, 2023, doi:10.3390/plants12010183_

Round 1
Reviewer 1 Report
Overall, the topic proposed by the manuscript is interesting and in line with journal aims and scope.
The article entitled "Organic versus non-organic plant-based foods - a comparative study on phenolic compounds composition and antioxidant capacity" presents the comparative results in terms of antioxidant capacity, total content of phenolics, anthocyanins, flavan-3-ols and flavonols for thirteen plant-based foods cultivated in organic and conventional systems.
The authors address a topic of novelty and interest both from the point of view of producers and from the perspective of population health. The nutritional differences between products obtained from plants grown in organic and conventional systems can constitute solid arguments for encouraging the large-scale cultivation of plants in organic systems.
The manuscript overall quality is good and can be published. The following changes should be done:
1. I believe that the specialized literature, regarding organic agriculture, can be improved by referring to more current bibliographic sources, see The word of organic agriculture
2. Please mention the sources (L59-60) Information supporting both agricultural practices can be obtained from the literature
3. The formulation of the hypotheses would bring increased value to the study.
4. Considering the results obtained, I suggest you clearly formulate the conclusions, if possible, for each analyzed plant.
5. Please mention the limitations of your study.
Author Response
Reviewer 1
Comments:
Overall, the topic proposed by the manuscript is interesting and in line with journal aims and scope.
The article entitled "Organic versus non-organic plant-based foods - a comparative study on phenolic compounds composition and antioxidant capacity" presents the comparative results in terms of antioxidant capacity, total content of phenolics, anthocyanins, flavan-3-ols and flavonols for thirteen plant-based foods cultivated in organic and conventional systems.
The authors address a topic of novelty and interest both from the point of view of producers and from the perspective of population health. The nutritional differences between products obtained from plants grown in organic and conventional systems can constitute solid arguments for encouraging the large-scale cultivation of plants in organic systems.
The manuscript overall quality is good and can be published. The following changes should be done:
- I believe that the specialized literature, regarding organic agriculture, can be improved by referring to more current bibliographic sources, see the world of organic agriculture
Regarding reviewer 1's comments, we are grateful for your suggestion. We have considered appropriate to add the following text to the introduction section, on page 1, line 35-37:
In fact, the organic agriculture is practiced in 187 countries and more than 106 billion euros were sold globally in 2019 in the organic food and drink sector [4].
- Please mention the sources (L59-60) Information supporting both agricultural practices can be obtained from the literature
Thank you for this observation. We have added references of studies supporting both organic and non-organic farming systems, on page 2, line 61:
Information supporting both agricultural practices can be obtained from the literature [6,13,14].
- The formulation of the hypotheses would bring increased value to the study
In consideration of reviewer 1's observation, we considerer to add the following hypotheses on page 2, line 63-66:
Based on the above, it is plausible to hypothesize that agriculture practices, for instance non-organic farming (NORG) and organic farming (ORG) generates a differential effect on phenolic synthesis and antioxidant activity among plant-based foods.
- Considering the results obtained, I suggest you clearly formulate the conclusions, if possible, for each analyzed plant.
Thank you for this suggestion. We have reformulated the conclusions and have included the following paragraph on page 11, line 453-455:
Indeed, orange and tomato T. were the only two species that showed differences across all parameters, the latter showing a clear trend of higher quantity in the NORG sample.
- Please mention the limitations of your study.
Thank you for this observation, we felt that a limitation of this study was that only the content of main phenolic subclasses was studied, without taking into account the individual concentration of phenolic compounds by LC-MS/MS. This could be studied in a further study. Therefore, the following paragraph has been included on page 11, lines 455-457:
However, this study has some limitations such as the lack of individual identification of phenolic compounds by chromatographic techniques and the limited information regarding the cultivar.

Reviewer 2 Report
The paper Cruz-Carrión et al. aimed to study influence of organic (ORG) or non-organic (NORG) farming on the phenolic content and ORAC value of selected cultures. Generally, the work looks a bit superficial due to a weak validity and the lack of the basic technical parameters.
It is not clear at all, how the cultures included in the experiment were chosen? Are these plants accumulators of something or something else? At the moment, the choice of research objects does not seem obvious and looks like a set of random plants. Have organically (ORG) and non-organically (NORG) culture systems been previously used for the selected objects? What were the results? Moreover, if these objects have already been studied before, then why are you decided to repeat experiment?
I do not understand the technical aspects of plant cultivation at all. Apparently, the authors were not harvested the cultures, but only purchased 2 kg of samples from different farms. Were they grown? in a greenhouse or open-field? And what exactly is an organically (ORG) and non-organically (NORG) cultivation system? Is there a standard protocol for harvesting? How did the authors control the cultivation conditions?
Have different chemicals and crop rotation systems been used to cultivate different plants or not? Can it be argued that one botanical specimen was cultivated in one farm? The importance is the same climatic conditions, soil, water regime and others. If there is a possibility that different conditions are used for one sample, then such samples cannot be compared as a pair. Moreover, different types of cultivation [organically (ORG) and non-organically (NORG)] for different plants can lead to different data, which also cannot be considered within the same study. Until the methodological part of the experiment is clarified, this study cannot be considered reliable.
Can the data obtained as a result of a six-month study be considered reliable? The data of 2-3 year experiments are usual for such studies.
Author Response
Reviewer 2
Comments:
The paper Cruz-Carrión et al. aimed to study influence of organic (ORG) or non-organic (NORG) farming on the phenolic content and ORAC value of selected cultures. Generally, the work looks a bit superficial due to a weak validity and the lack of the basic technical parameters.
It is not clear at all, how the cultures included in the experiment were chosen? Are these plants accumulators of something or something else? At the moment, the choice of research objects does not seem obvious and looks like a set of random plants. Have organically (ORG) and non-organically (NORG) culture systems been previously used for the selected objects? What were the results? Moreover, if these objects have already been studied before, then why are you decided to repeat experiment?
Regarding reviewer 2's comments, this study is part of the local project focused on the study of local products in order to increase scientific data about the benefits of local consumption. For this, our group had conducted several studies to evidence the health effects of the consumption of local plant-based foods. One of the goals of this research was to determine the distinctive phenolic signature of the main plant-based foods grown into this area. For this purpose, approximately 2 kg of plant-based foods from different farming systems and cultivars were donated by farmers. A total of 124 samples were collected and analysed for phenolic content and a database with these results was generated. From these sample, we selected those that were produced both under organic and non-organic farming system in order to determine differences in the content of phenolic compounds and antioxidant activity. Therefore, we are not repeating any experiment, actually we are producing data that will be useful for the local producers.
I do not understand the technical aspects of plant cultivation at all. Apparently, the authors were not harvested the cultures, but only purchased 2 kg of samples from different farms. Were they grown? in a greenhouse or open-field? And what exactly is an organically (ORG) and non-organically (NORG) cultivation system? Is there a standard protocol for harvesting? How did the authors control the cultivation conditions?
With regard to reviewer 2's comments, we are very pleased to expand and deepen these questions.
The cultures were donated by farmers of this area, who were in charge of its harvesting, according to standard protocol provided by us, which basically emphasized that the products should be harvested at commercial maturity and taken from different plants and parts of the growing area [1]. The following statement has been added on page 10, lines 373-376:
The cultures were harvested as previously described [30]. According to the information provided by them, these crops were cultivated in open-field. ORG cultivation system meets the certification provided by the Catalan Council of the Organic Production.
Unfortunately, we were not able to control the cultivation conditions, since our work focused on the influence of the organic or non-organic cultivation system on the synthesis and accumulation of phenolic compounds. However, we consider that it would be very interesting that the cultivation conditions are considered in a future study.
Have different chemicals and crop rotation systems been used to cultivate different plants or not?
According to the information provided by farmers, the fertilizers, soil conditioners and nutrients used in organic production comply with the provisions of the current European regulations [2]. The following statement had been included on page 10, lines 371-372:
indeed, the fertilizers, soil conditioners and nutrients used in organic production comply with the provisions of the current European regulations [2].
Can it be argued that one botanical specimen was cultivated in one farm?
All plants were harvested between May 2018 to February 2019 and donated by farmers and agricultural companies from Camp de Tarragona – Spain, but we do not have the exact information about the specific details of each botanical. Actually, this is one of the limitations of the study that we have include in the conclusions sections as suggested by the reviewer 1.
The importance is the same climatic conditions, soil, water regime and others. If there is a possibility that different conditions are used for one sample, then such samples cannot be compared as a pair. Moreover, different types of cultivation [organically (ORG) and non-organically (NORG)] for different plants can lead to different data, which also cannot be considered within the same study. Until the methodological part of the experiment is clarified, this study cannot be considered reliable.
Regarding reviewer 2's comments, agrometeorological parameters such as: average daily relative humidity (%), average daily temperature (°C), scalar wind speed at 2m (m/s), accumulated daily precipitation (mm/m²), daily global solar radiation (MJ/m²), reference evapotranspiration (mm/m²), and altitude (m) were collected [3] and some statics analyse were performed, without finding any significant correlation or significant effect with phenolic content or antioxidant activity. Therefore, we consider it appropriate to include the following sentence on page 8, line 274-277:
In our work, agrometeorological parameters such as: relative humidity, daily temperature, scalar wind speed, accumulated daily precipitation, daily global solar radiation, reference evapotranspiration and altitude were collected and analysed, without finding any significant correlation or significant effect on phenolic content or antioxidant activity (supplementary table S1).
Can the data obtained as a result of a six-month study be considered reliable? The data of 2-3 year experiments are usual for such studies.
In response to reviewer 2's comments, the results obtained reflect the phenolic composition of a group of plant-based foods that have been harvested in the different seasons of the year. In this line, a work by Young et al. [4] studied the phenolic levels in leaf lettuce (Lactuca sativa L. cv. Kalura and Red Sails), collards (Brassica oleracea L. cv. Top Bunch), and pac choi (Brassica rapa L. cv. Mei Qing) cultivated under organic and conventional production systems during the summer 2003. Another study evaluated the phenolic acids concentration in wheat harvested in spring and winter 2008 [5]. Likewise, Ponder and Hallmann [6] studied the effects of ORG and NORG farming systems on the polyphenol content in three raspberry cultivars collected in the summer 2013 and in the autumn 2014. Therefore, we consider that the information obtained in our analysis could be of interest of the readers of Plants. However, we are aware that this is a limitation of the study and we have discussed this in the discussion section, on page 7, lines 220-228:
The results obtained reflect the phenolic content of a group of plant-based foods that have been harvested in the different seasons of the year (Spring 2018 – Winter 2019). In this line, a work by Young et al. [4] studied the phenolic levels in leaf lettuce (Lactuca sativa L. cv. Kalura and Red Sails), collards (Brassica oleracea L. cv. Top Bunch), and pac choi (Brassica rapa L. cv. Mei Qing) cultivated under ORG and NORG during the summer 2003. Another study evaluated the phenolic acids concentration in wheat harvested in spring and winter 2008 [5]. Likewise, Ponder and Hallmann [6] studied the effects of ORG and NORG farming systems on the polyphenol content in three raspberry cultivars collected in the summer 2013 and in the autumn 2014.

Round 2
Reviewer 2 Report
The authors worked to rectify their mistakes and the final vertion of the manuscript looks much better. The paper may be accepted in pesent form
Author Response
Thank you for your comments and revision so far.